# S-nitrosothiol homeostasis maintained by ADH5 facilitates STING-dependent host defense against pathogens

Mutian Jia [1,2,5], Li Chai [1,2,5], Jie Wang[1,2], Mengge Wang[1,2], Danhui Qin[1,2], Hui Song[1,2], Yue Fu[1,3], Chunyuan Zhao[4], Chengjiang Gao [1,2], Jihui Jia[1] & Wei Zhao [1,2] ✉

Oxidative (or respiratory) burst confers host defense against pathogens by generating reactive species, including reactive nitrogen species (RNS). The microbial infection-induced excessive RNS damages many biological molecules via S-nitrosothiol (SNO) accumulation. However, the mechanism by which the host enables innate immunity activation during oxidative burst remains largely unknown. Here, we demonstrate that S-nitrosoglutathione (GSNO), the main endogenous SNO, attenuates innate immune responses against herpes simplex virus-1 (HSV-1) and *Listeria monocytogenes* infections. Mechanistically, GSNO induces the S-nitrosylation of stimulator of interferon genes (STING) at Cys257, inhibiting its binding to the second messenger cyclic guanosine monophosphate-adenosine monophosphate (cGAMP). Alcohol dehydrogenase 5 (ADH5), the key enzyme that metabolizes GSNO to decrease cellular SNOs, facilitates STING activation by inhibiting S-nitrosylation. Concordantly, *Adh5* deficiency show defective STING-dependent immune responses upon microbial challenge and facilitates viral replication. Thus, cellular oxidative burst-induced RNS attenuates the STING-mediated innate immune responses to microbial infection, while ADH5 licenses STING activation by maintaining cellular SNO homeostasis.

Stimulator of interferon genes (STING) is a crucial element in the innate immune system that recognizes the second messenger cyclic guanosine monophosphate (GMP)-adenosine monophosphate (AMP) (cGAMP) produced by the cytosolic DNA sensor, cGAMP synthase (cGAS)[1–7]. Within this pathway, cGAS senses microbial and host-derived DNA in the cytoplasm and synthesizes cGAMP, which binds to dimeric STING at the endoplasmic reticulum (ER). cGAMP binding promotes the dimerization and polymerization of STING. It induces STING

trafficking from the ER to the ER-Golgi intermediate compartment (ERGIC) and Golgi apparatus, where it recruits TANK-binding kinase 1 (TBK1) and interferon (IFN) regulatory factor 3 (IRF3), leading to the expression of type I IFNs[3–5,8,9]. In addition to inducing type I IFNs production, STING activates NF-κB to initiate inflammation in a TBK1-independent manner[10]. Optimal activation of STING is vital for host defense against pathogens and maintaining immune homeostasis. Aberrant STING activity has been implicated in various diseases,

[1]Department of Pathogenic Biology, Key Laboratory for Experimental Teratology of the Chinese Ministry of Education, and Key Laboratory of Infection and Immunity of Shandong Province, School of Basic Medical Science, Qilu Hospital, Cheeloo College of Medicine, Shandong University, Jinan, Shandong, China. [2]State Key Laboratory of Microbial Technology, Shandong University, Jinan, Shandong, China. [3]Department of Physiology & Pathophysiology, School of Basic Medical Science, Cheeloo College of Medicine, Shandong University, Jinan, Shandong, China. [4]Department of Cell Biology, School of Basic Medical Science, Cheeloo College of Medicine, Shandong University, Jinan, Shandong, China. [5]These authors contributed equally: Mutian Jia, Li Chai. ✉e-mail: wzhao@sdu.edu.cn

including infections, autoimmune and inflammatory disorders, and cancer[3]. Therefore, STING activity requires tight control to facilitate the elimination of pathogens and avoid detrimental effects.

Oxidative (or respiratory) burst is a vital host defense mechanism that involves the rapid generation and release of reactive oxygen species (ROS) and reactive nitrogen species (RNS), which contribute to the elimination of microorganisms[11,12]. ROS and RNS belong to a family of reactive species derived from oxygen or nitric oxide (NO), respectively, that modify the redox-sensitive residues of target proteins and therefore provide redox switches between active and inactive states of proteins[13]. The function of NO is primarily mediated by S-nitrosation/S-nitrosylation, a covalent modification of thiol groups on small molecules (such as glutathione [GSH]) or cysteine residues of proteins by NO or its derivatives to form S-nitrosothiols (SNOs)[13]. Cellular SNO homeostasis is tightly controlled and perturbations in homeostasis lead to an altered redox state, which is associated with a plethora of pathological conditions[12,13]. Previously, we reported that viral infection causes excessive production of toxic lipid peroxides (lipid ROS, such as 4-HNE and MDA), which attenuates antiviral innate immunity by promoting STING carbonylation and facilitates the immune escape of viruses[14]. However, the potential effects of the oxidative burst, especially how excessive SNO regulates STING activation and the mechanism by which the host enables STING activation during oxidative burst, remain unclear.

S-nitrosoglutathione (GSNO) is the main endogenous low molecular weight SNO and serves as a stable endogenous reservoir providing a source of bioavailable NO[15,16]. GSNO transfers its NO moiety to a cysteine thiol, resulting in a post-translational modification (PTM) termed protein S-nitrosylation[17]. Alcohol dehydrogenase 5 (ADH5), also known as GSNO reductase (GSNOR), is the key enzyme that metabolizes GSNO to decrease the total levels of SNO[15,16]. ADH5 dysregulation alters cellular SNO homeostasis and S-nitrosylation to affect the function of Th17, B, and cancer cells[18–20]. Here, we demonstrate that ADH5 enhances the cGAS-STING pathway in macrophages by suppressing the S-nitrosylation of STING. GSNO attenuated STING activation and type I IFN expression induced by herpes simplex virus-1 (HSV-1) and *Listeria monocytogenes* (*L. monocytogenes*) infection. Mechanistically, GSNO promoted the S-nitrosylation of STING at Cys257 to inhibit its binding to cGAMP. Accordingly, *Adh5* deficiency suppressed STING-dependent antiviral innate responses and facilitated viral replication. Our results reveal the mechanism by which oxidative burst, the physiological process that occurs during pathogen infection, regulates innate immune responses and suggest that SNO homeostasis maintained by ADH5 is required for STING activation. Our study uncovered a novel PTM that controls STING activity and suggests that GSNO and ADH5 are priming targets for the intervention of diseases caused by abnormal STING activation.

## Results

### GSNO inhibits cGAS-STING activation

Pathogens infection causes the robust generation and release of ROS and RNS (Supplementary Fig. 1a–d), which is called as oxidative (or respiratory) burst. S-nitrosothiols (SNOs) are the main kinds of RNS produced by NO covalent modification of thiol groups[13]. To investigate the potential role of SNO homeostasis in the DNA sensing pathway, we first examined the effects of NO donors on cGAS-STING activation in mouse peritoneal macrophages (PMs). GSNO and S-nitroso-N-acetyl-DL-penicillamine (SNAP, an NO donor) both dose-dependently attenuated HSV-1 (a DNA virus recognized by cGAS) infection-induced *Ifnb* mRNA expression (Fig. 1a, b), but did not influence the viability of PMs (Supplementary Fig. 1e, f). Similarly, GSNO and SNAP markedly inhibited HSV-1 infection-induced IFN-β and interleukin (IL)−6 secretion (Fig. 1c and Supplementary Fig. 1g). Furthermore, GSNO and SNAP both attenuated IFN-stimulating DNA (ISD; recognized by cGAS), cGAMP, and 5,6-dimethylxanthenone-4-acetic acid (DMXAA; STING agonist)-induced IFN-β expression (Fig. 1c and Supplementary Fig. 1h–j). Phosphorylation of TBK1 and IRF3 is a critical step in the expression of type I

IFNs and STAT1, downstream of IFNs, controls the expression of IFN-stimulated genes (*Isgs*). GSNO and SNAP suppressed HSV-1 infection-, ISD-, cGAMP-, and DMXAA-induced phosphorylation of TBK1, IRF3, and STAT1 (Fig. 1d–f and Supplementary Fig. 1k–o). Consequently, cGAMP-induced mRNA expression of *Isgs* (including *Isg15*, *Isg54*, *Isg56*, and *Mx-1*) and *Cxcl10* (an IRF3-dependent chemokine) was also suppressed by both GSNO and SNAP (Fig. 1g and Supplementary Fig. 1p–r). Similarly, GSNO inhibited HSV-1 infection- and cGAMP-induced *IFNB* mRNA expression (Fig. 1h), and ISD-induced phosphorylation of IRF3 and STAT1 in human monocyte THP-1 cells. (Fig. 1i). Both GSNO and SNAP attenuated IFN-β luciferase activation in DMXAA-stimulated 293-Dual hSTING-A162 cells (Fig. 1j).

*L. monocytogenes* infection, which releases double-stranded DNA intracellularly, induces IFN-β expression in a STING-dependent manner[21]. GSNO and SNAP both attenuated *L. monocytogenes* infection-induced production of IFN-β, *Cxcl10* and *Isgs* (Fig. 1k and Supplementary Fig. 1s, t, v) and the phosphorylation of TBK1, IRF3, and STAT1 in PMs (Fig. 1l and Supplementary Fig. 1u). Overall, these data indicated that GSNO and SNAP both inhibited the cGAS-STING pathway and decreased IFN-β expression.

### GSNO attenuates host defenses against HSV-1 infection

Type I IFNs and *Isgs* are crucial for eliminating viral invasion in the host. Consistent with their inhibitory roles in IFN-β and *Isgs* expression, GSNO, and SNAP both enhanced HSV-1 replication in PMs (Fig. 2a–c). We investigated the regulatory effects of GSNO on host defense in the context of HSV-1 infection in vivo. GSNO inhibited serum levels of IFN-β and IL−6 in HSV-1 infected mice (Fig. 2d, e). *Ifnb* and *Cxcl10* mRNA expression in the spleen, brain, and lung tissues of GSNO-treated mice was lower than that in the controls (Fig. 2f–h and Supplementary Fig. 2). Accordingly, HSV-1 replication in the lungs of GSNO-treated mice was higher than that in control mice (Fig. 2i, j). Severe infiltration of immune cells was observed in the lungs of GSNO-treated mice following HSV-1 infection compared with that in control mice (Fig. 2k). Moreover, GSNO-treated mice were more susceptible to HSV-1 infection than the control mice (Fig. 2l). These data indicate that GSNO attenuates HSV-1 infection-induced IFN-β expression and thus promotes HSV-1 replication both in vitro and in vivo.

### GSNO induces S-nitrosylation of STING to inhibit cGAMP binding

To clarify the mechanisms by which GSNO attenuates the cGAS-STING pathway, we examined the effects of GSNO on cGAS activity. GSNO and SNAP had no effect on HSV-1 infection-induced cGAMP production (Fig. 3a, b). Next, we examined the effect of GSNO on STING activation. GSNO markedly inhibited the binding of cGAMP to STING compared with that in the control (Fig. 3c). Consequently, STING dimerization was dramatically suppressed by GSNO or SNAP treatment compared with that in the control (Fig. 3d and Supplementary Fig. 3a, b). GSNO or SNAP treatment inhibited HSV-1 infection- and cGAMP-induced STING oligomerization (Fig. 3e, f). cGAMP-induced localization of STING in the Golgi was also blocked by GSNO treatment compared with that in the control (Fig. 3g, h). Additionally, GSNO inhibited HSV-1 infection-induced STING phosphorylation compared with that in the control (Supplementary Fig. 3c). To clarify the effects of GSNO on RLR signaling, we performed additional experiments. GSNO treatment had no effects on VSV- (an RNA virus recognized by RIG-I) and poly(I:C) transfection-induced IFN-β expression in both wild-type and *Sting1* deficient macrophages (Fig. 3i, j). Similarly, no difference in VSV- and poly(I:C)-induced IFN-β expression was observed in GSNO-treated THP-1 cells (Fig. 3k). Overall, these data indicate that GSNO targets STING and prevents its binding to cGAMP.

GSNO transfers an NO moiety to proteins to induce S-nitrosylation[15,16]. Thus, we then investigated whether STING is S-nitrosylated during pathogen infection using the irreversible

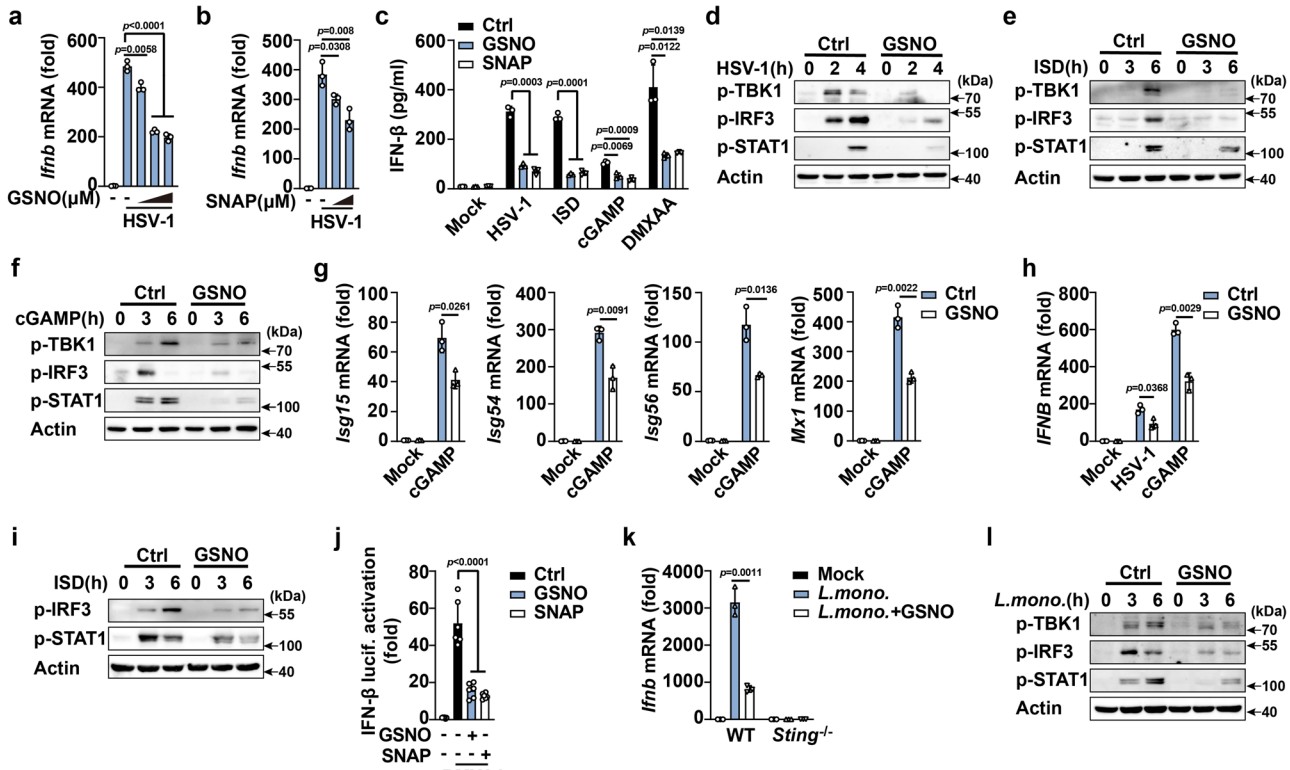

**Fig. 1 | GSNO inhibits cGAS-STING activation.** Quantitative polymerase chain reaction (qPCR) analysis of *Ifnb* mRNA expression in mouse peritoneal macrophages (PMs) pretreated with increasing concentrations of GSNO (0, 100, 250, and 500 μM) (**a**) or SNAP (0, 250, and 500 μM) (**b**), following herpes simplex virus-1 (HSV-1) infection. **c** Enzyme-linked immunosorbent assay (ELISA) analysis of interferon (IFN)-β secretion in mouse PMs pretreated with solvent (Ctrl), GSNO, or SNAP, plus stimulation as indicated. **d**–**f** Immunoblot assays of p-TBK1, p-IRF3, and p-STAT1 in PMs pretreated with solvent (Ctrl) or GSNO and then stimulated with HSV-1, IFN-stimulating DNA (ISD), or cyclic guanosine monophosphate (GMP)-adenosine monophosphate (AMP) (cGAMP). **g** qPCR analysis of *Isg15, Isg54, Isg56*, and *Mx1* mRNA expression in PMs pretreated with solvent (Ctrl) or GSNO and then stimulated with cGAMP. **h** qPCR analysis of *IFNB* mRNA expression in THP1 cells

pretreated with solvent (Ctrl) or GSNO and then stimulated with HSV-1 or cGAMP. **i** Immunoblot assays of p-IRF3 and p-STAT1 in THP1 cells pretreated with solvent (Ctrl) or GSNO and then stimulated with ISD. **j** Luciferase activity assays of IFN-β activation in 293-Dual hSTING-A162 cells pretreated with solvent (Ctrl), GSNO, or SNAP, following stimulation by DMXAA. **k** qPCR analysis of *Ifnb* mRNA expression in PMs from *Sting^+/+* or *Sting^-/-* mice pretreated with solvent (Ctrl) or GSNO and then infected with *L. monocytogenes*. **l** Immunoblot assays of p-TBK1, p-IRF3, and p-STAT1 in PMs pretreated with solvent (Ctrl) or GSNO and then infected with *L. monocytogenes*. Data represent mean ± standard deviation (SD) or one representative image from three independent experiments. The *p* values were calculated using unpaired two-sided *t* test and adjustments were made for multiple comparisons.

biotinylation procedure (IBP)[22]. HSV-1-and *L. monocytogenes*-infection induced S-nitrosylation of endogenous STING in PMs, but not cGAS, IRF3 or TBK1 (Fig. 4a, b). Similarly, STING was S-nitrosylated in STING-FLAG overexpressed human embryonic kidney (HEK293T) cells, and GSNO treatment enhanced STING S-nitrosylation compared with that in control (Fig. 4c). STING S-nitrosylation was confirmed by an in vitro S-nitrosylation assay using the recombinant human STING protein (Fig. 4d). Three residues were predicted as S-nitrosylation sites[23], and residues Cys257 and Cys309 were conserved across species (Supplementary Fig. 4a, b). To further evaluate the potential S-nitrosylation of STING, we performed liquid chromatography-mass spectrometry (LC-MS) analysis using recombinant human STING protein in an in vitro S-nitrosylation assay. Residue Cys257, but not Cys309, was identified as a potential S-nitrosylation site for STING (Fig. 4e and Supplementary Fig. 4c). Next, we constructed STING C257S and C309S mutants, in which a Cys (C)-to-Ser (S) point mutation was introduced. STING S-nitrosylation was significantly reduced in HEK293T cells transfected with STING C257S but not with C309S (Fig. 4f). The STING C257S mutant inhibited cGAMP binding to STING in HEK293T cells (Fig. 4g), suggesting that Cys257 is vital for the binding between cGAMP and STING. Based on the crystal structure of STING in complex with cGAMP[24], the Cys257 residue is in close proximity to the cGAMP binding sites, including Glu260 and Thr263 (Protein Data Bank ID code: 4KSY). To clarify whether S-nitrosylation of STING at Cys257

affects cGAMP binding, we simulated S-nitrosylation of STING at Cys257 (STING-C257-NO) and compared its binding activity with that of wild-type STING using molecular docking and dynamics simulations (Supplementary Fig. 4d). Compared with that of wild-type STING, the hydrogen bonds between Glu260/Thr263 and cGAMP were abolished in STING-C257-NO (Fig. 4h, i). Furthermore, the binding energy between cGAMP and STING-C257-NO was reduced compared with that of wild-type STING after 80 ns of simulation (Supplementary Fig. 4e). Moreover, the binding sites of STING and other ligands (such as c-di-GMP) are very similar to those of cGAMP, and Glu260/Thr263 is also required to form hydrogen bonds with c-di-GMP (Protein Data Bank ID code: 4EF4) (Supplementary Fig. 4f). GSNO treatment also inhibited c-di-GMP-induced phosphorylation of TBK1, IRF3, STAT1 and expression of *Ifnb* (Supplementary Fig. 4g, h). STING-R238A/Y240A mutant is incapable of binding to cGAMP, but S-nitrosylation of STING was also observed in STING-R238A/Y240A mutant, suggesting that R238/Y240 are not essential residues for S-nitrosylation of STING (Fig. 4j). Collectively, these data indicate that GSNO induces the S-nitrosylation of STING at C257 to inhibit cGAMP binding.

## ADH5 attenuates S-nitrosylation of STING to facilitate its activation

ADH5 is the critical enzyme that metabolizes GSNO to decrease cellular SNO levels. *Adh5* deficiency increases both cellular levels

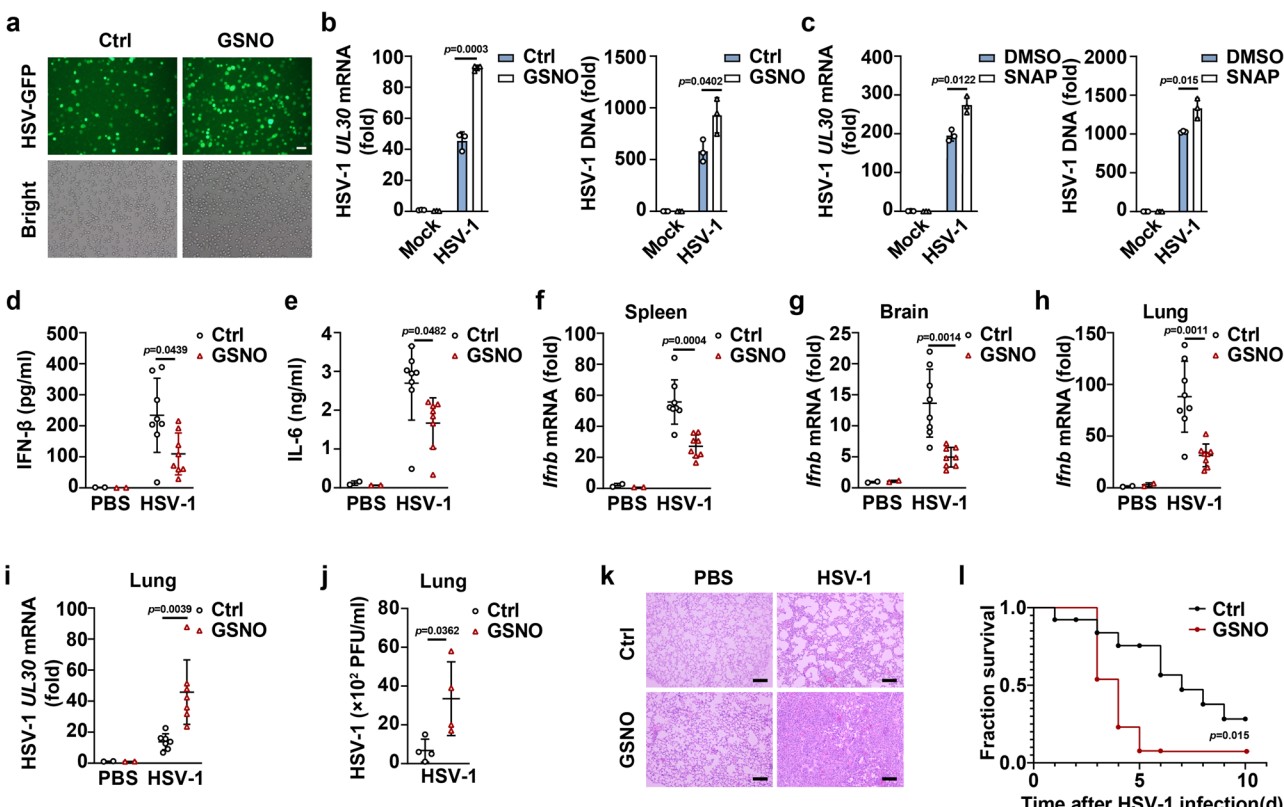

**Fig. 2 | GSNO attenuates host defenses against HSV-1 infection. a** Microscopy analysis of HSV-1 replication in PMs pretreated with solvent (Ctrl) or GSNO, and then infected with HSV-GFP for 12 h. Scale bars, 100 μm. qPCR analysis of HSV-1 *UL30* mRNA level (left) and HSV-1 DNA level (right) in PMs pretreated with solvent (Ctrl), GSNO (**b**), or SNAP (**c**), and then infected with HSV-1. **d**–**l** C57BL/6 J mice were pretreated with solvent (Ctrl) or GSNO and then infected with HSV-1. Serum levels of IFN-β (**d**) and IL-6 (**e**) were analyzed by ELISA (PBS group, *n* = 2; HSV-1 group, *n* = 8). qPCR analysis of *Ifnb* mRNA expression in the spleen (**f**), brain (**g**), and lung (**h**) tissues (PBS group, *n* = 2; HSV-1 group, *n* = 8). The HSV-1 viral burden was determined by measurement of HSV-1 *UL30* mRNA levels in lung tissues (**i**) (PBS group, *n* = 2; HSV-1 group, *n* = 7). Plaque analysis of homogenizes of lung (**j**) from C57BL/6 J mice were pretreated with solvent (Ctrl) or GSNO (*n* = 4), and then infected with HSV-1 for 2 days. Hematoxylin and eosin staining of lung tissue sections. Scale bar: 200 μm (**k**). The Kaplan-Meier method was used to evaluate survival curves (*n* = 13 per group) (**l**). Statistical significance was determined by unpaired two-sided multiple Student's *t* tests in (**a**–**k**) or the log-rank Mantel-Cox test in (**l**). Data represent mean ± SD or one representative image from three independent experiments.

of GSNO and total protein S-nitrosylation[25]. We investigated the potential role of ADH5 in regulating the S-nitrosylation of STING. *Adh5* deficiency enhanced S-nitrosylation of STING in HSV-1-infected PMs and inhibited cGAMP-triggered dimerization of STING compared with that in the controls (Fig. 5a, b and Supplementary Fig. 5a). Consequently, *Adh5* deficiency inhibited HSV-1-, ISD-, cGAMP-, and DMXAA-induced phosphorylation of TBK1, IRF3, and STAT1, and the expression of IFN-β compared with that in the control (Fig. 5c–e and Supplementary Fig. 5b–d). HSV-1 infection-induced *Cxcl10* mRNA expression was attenuated by *Adh5* deficiency (Supplementary Fig. 5e). Consistently, *Adh5* deficiency enhanced HSV-1 replication in PMs (Fig. 7b, c). Similarly, *Adh5* knockdown significantly suppressed HSV-1- and DMXAA-induced phosphorylation of TBK1, IRF3, and STAT1, as well as HSV-1-, ISD-, cGAMP-, and DMXAA-induced mRNA expression of *Ifnb* (Fig. 5f, g and Supplementary Fig. 5f, g). N6022, a selective ADH5 inhibitor[26], dose-dependently inhibited the phosphorylation of IRF3 and STAT1 and decreased IFN-β and CXCL10 expression in HSV-1-infected PMs compared with that in the control (Fig. 5h, i and Supplementary Fig. 5h–j). ADH5 enhanced wild-type STING-induced IFN-β luciferase activation, but not the STING C257S mutant (Fig. 5j). Next, we investigated the physiological relevance of the effects of ADH5 on STING activation in vivo. The serum level of IFN-β and IL-6 secretion induced by HSV-1 infections in *Adh5*-deficient mice was much lower than that in wild-type mice (Fig. 5k, l). Furthermore, the serum level of IFN-β secretion induced by DMXAA injection in *Adh5*-deficient mice was much lower than that in wild-

type mice (Fig. 5m). DMXAA injection-induced *Ifnb* mRNA expression in the spleen and lungs was also suppressed by *Adh5* deficiency (Fig. 5n). These data collectively indicate that ADH5 attenuates the S-nitrosylation of STING and enhances STING activation both in vitro and in vivo.

## ADH5 is downregulated during pathogens infection that attenuates host innate responses

STING-driven type I IFN responses are crucial host defense mechanisms against pathogens. HSV-1 and *L. monocytogenes* infection markedly decreased ADH5 expression, whereas IFN-β stimulation enhanced ADH5 expression in PMs (Fig. 6a–f). These results suggest that ADH5 is downregulated during pathogens infection to manipulate host defenses. Pathogens have evolved a variety of strategies to circumvent host defense, such as controlling DNA methylation[27]. The promoter region of both human and mouse *ADH5* revealed the presence of CpG islands (Supplementary Fig. 6), which serves as a predictive marker of epigenetic silencing by DNA methylation. Analysis of scMethBank, a database of single-cell methylation maps for human and mouse[28], showed that both mouse and human ADH5 is methylated in a variety of cell types. It has been reported that the expression of ADH5 is regulated by DNA methylation[29]. Azacitidine is a specific inhibitor to suppress DNA methylation, which is widely used to prove the correlation between loss of methylation in specific gene regions and activation of the related genes[30,31]. We found that pathogens infection lost the inhibitory effects on ADH5 expression in Azacitidine

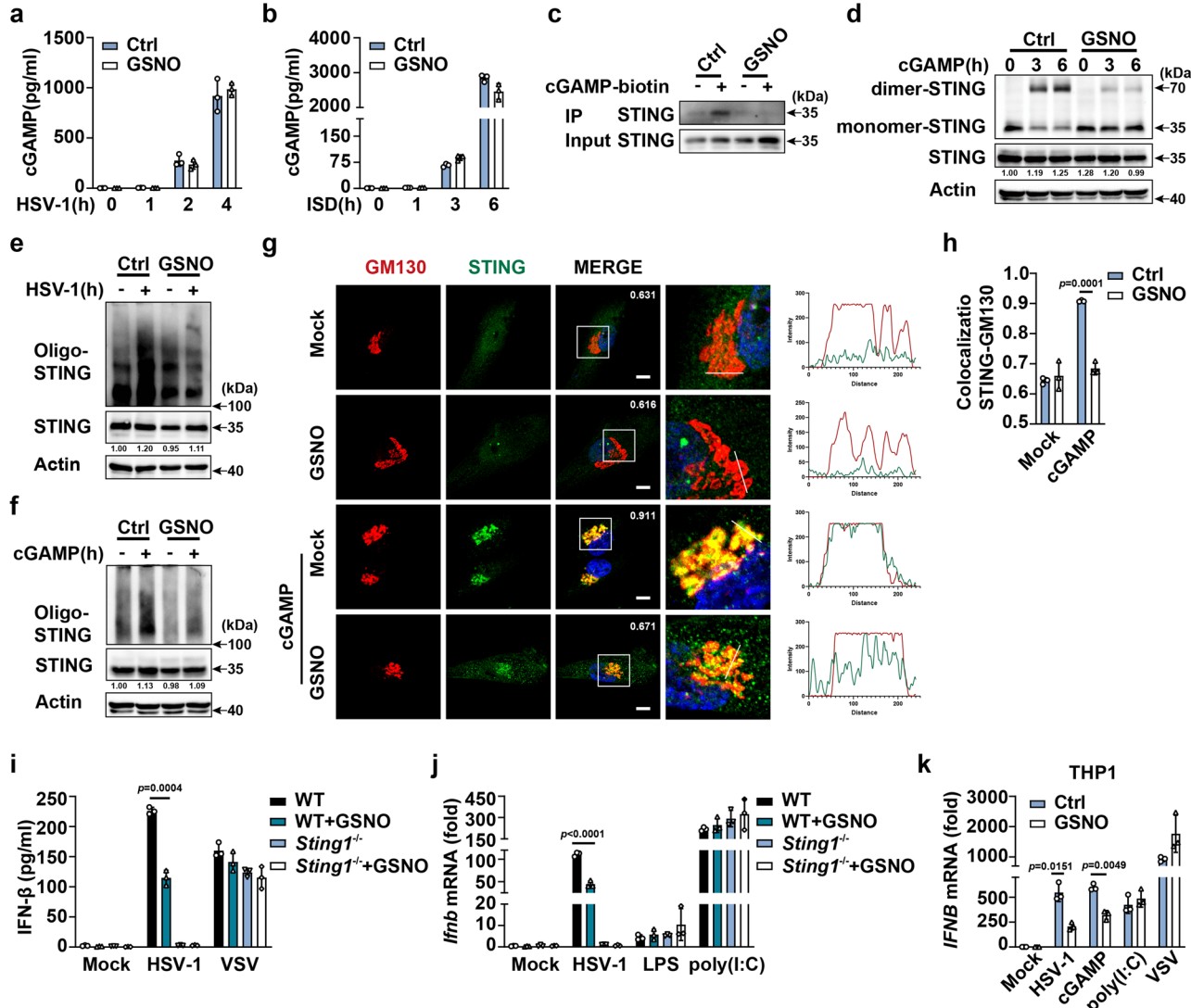

**Fig. 3 | GSNO targets STING.** ELISA analysis of cGAMP production in HSV-1-infected (**a**) and ISD-stimulated (**b**) PMs pretreated with solvent (Ctrl) or GSNO. **c** Co-immunoprecipitation analysis of cGAMP-biotin binding to STING in HSV-1-infected PMs pretreated with solvent (Ctrl) or GSNO. **d** Immunoblot assays of dimer-STING in cGAMP-stimulated PMs pretreated with solvent (Ctrl) or GSNO. Immunoblot assays of Oligo-STING in HSV-1-infected (**e**) and cGAMP-stimulated (**f**) PMs pretreated with solvent (Ctrl) or GSNO. **g** Confocal microscopy analysis of the co-localization of STING (green) and cis-Golgi (GM130, red) in BJ cells pretreated with solvent (Ctrl) or GSNO and then stimulated with cGAMP. The intensity profiles of each line were quantified by Image J software. **h** Colocalization analysis of STING and cis-Golgi (GM130) by Manders' Colocalization Coefficients (MCC). **i** ELISA analysis of interferon (IFN)-β secretion in PMs from *Sting1*^+/+ or *Sting1*^-/- mice pretreated with solvent (Ctrl) or GSNO, plus stimulation as indicated. **j** qPCR analysis of *Ifnb* mRNA expression in PMs from *Sting1*^+/+ or *Sting1*^-/- mice pretreated with solvent (Ctrl) or GSNO and then stimulated with HSV-1, LPS or poly(I:C). **k** qPCR analysis of *IFNB* mRNA expression in THP1 cells pretreated with solvent (Ctrl) or GSNO and then stimulated with HSV-1, cGAMP, poly(I:C) or VSV. Data represent mean ± SD or one representative image from three independent experiments. The *p* values were calculated using unpaired two-sided *t* test and adjustments were made for multiple comparisons.

treated macrophages (Fig. 6g, h). These results suggested that ADH5 is downregulated during pathogens infection by facilitating DNA methylation.

Next, we investigated the effects of ADH5 on innate immune responses against pathogens. HSV-1 infection-induced *Isgs* (including *Isg15*, *Isg54*, *Isg56*, and *Mx-1*) expression was suppressed by *Adh5* deficiency (Fig. 7a), while *Adh5* deficiency enhanced HSV-1 replication (Fig. 7b, c) compared with that in the control. In the context of *L. monocytogenes* infection, *Adh5* deficiency inhibited the phosphorylation of STING, TBK1, IRF3, and STAT1, and the expression of IFN-β, CXCL10, and *Isgs* in PMs (Fig. 7d–h). Similar results were observed in *Adh5*-knockdown PMs (Supplementary Fig. 7a, b). These data indicate that ADH5 is downregulated during pathogens infection to attenuate host innate immune responses (see the model in Supplementary Fig. 8).

## Discussion

The phagocyte oxidative burst confers host defense against a broad spectrum of viral and bacterial pathogens by generating reactive species, including ROS and RNS. As the products of cellular metabolism, reactive species in low and moderate concentrations show beneficial effects on host defense against infectious agents and protection of multiple cellular signaling systems[32,33]. However, excessive reactive species (termed oxidative and nitrosative stress) leads to detrimental effects by inducing the oxidation of proteins, altering their functions, and damaging many biological molecules[34,35]. For example, excess mitochondrial ROS causes pathogenic necrosis in mycobacterium-infected macrophages[36]. Moreover, excessive ROS induces oxidative deterioration of lipids (lipid peroxidation) to trigger ferroptosis in various cell types[37], such as CD8+ T cells, follicular helper T cells, and neutrophils[38–40]. Excess ROS in activated T cells trigger an oxidative

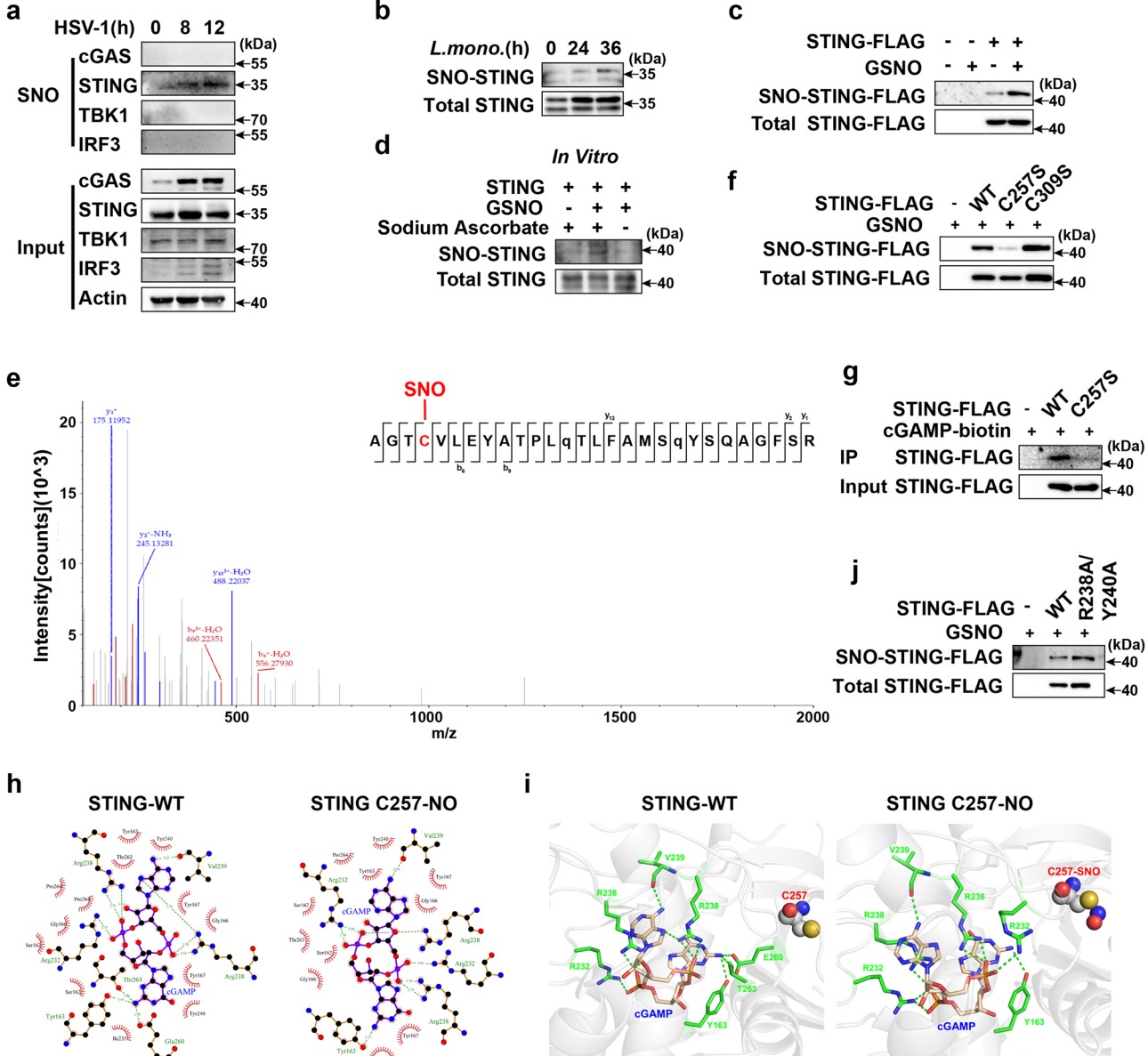

**Fig. 4 | GSNO induces S-nitrosylation of STING to inhibit cGAMP binding.**
**a**, **b** Immunoblot assays of STING S-nitrosylation in PMs infected with HSV-1 or *L. monocytogenes* using the irreversible biotinylation procedure (IBP). **c** Immunoblot assays of STING S-nitrosylation in HEK293T cells transfected with STING-FLAG and treated with solvent (Ctrl) or GSNO. **d** Immunoblot assays of in vitro S-nitrosylation of recombinant human STING protein treated with GSNO. **e** Liquid chromatography-mass spectrometry (LC-MS) spectra of STING S-nitrosylation at residue C257. **f** Immunoblot assays of S-nitrosylation in HEK293T cells transfected with empty vector, wild-type (WT) STING, or STING mutants (C257S or C309S)

using the IBP. **g** Co-immunoprecipitation analysis of cGAMP-biotin binding to STING in HEK293T cells transfected with empty vector, WT, or C257S STING mutant. **h**, **i** The binding pattern of cGAMP with WT STING or S-nitrosylation of STING at C257 (STING-C257-NO) after an 80–100 ns simulation. The dashed green lines represent hydrogen bonds, and the red gear represents hydrophobicity. **j** Immunoblot assays of S-nitrosylation in HEK293T cells transfected with empty vector, wild-type (WT) STING, or STING mutants (R238A/Y240A) using the IBP. Data are shown as one representative image from three independent experiments.

stress response that leads to translation repression and inhibition of T cell expansion[41]. Excessive RNS induces the S-nitrosylation of multiple biomolecules, which alters their structure and function via SNO accumulation[25,42–44]. Dysregulated S-nitrosylation under nitrosative stress has been implicated in various disease states, including immune, central nervous, and cardiovascular system disorders, as well as cancer[45–47]. Therefore, cellular SNO homeostasis is crucial for limiting aberrant S-nitrosylation and maintaining cellular function. GSNO, a type of RNS formed from NO and cellular thiol GSH under aerobic conditions, serves as a stable reserve for SNO[15]. In this study, we demonstrated that GSNO attenuated the cGAS-STING pathway by

inducing S-nitrosylation of STING at Cys257 to inhibit cGAMP binding, resulting in the suppression of type I IFN responses in HSV-1 and *L. monocytogenes* infections. The GSNO reductase ADH5 suppressed S-nitrosylation and activated STING. During pathogen infection, phagocyte oxidative burst releases ROS and RNS to eradicate invading pathogens. However, excessive ROS and RNS damage STING by inducing carbonylation and S-nitrosylation, respectively, which suppresses host innate responses. ADH5, together with glutathione peroxidase 4, maintain redox homeostasis by reducing cellular GSNO and lipid peroxidation to facilitate STING activation and enable host immunity against pathogens.

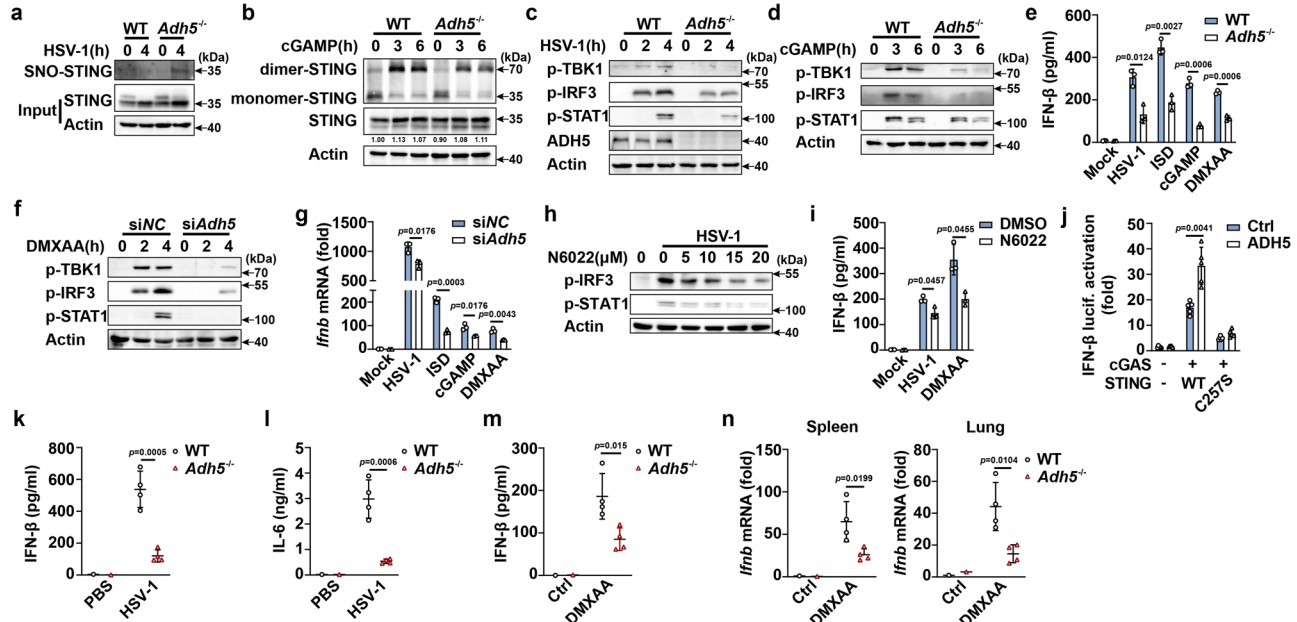

**Fig. 5 | ADH5 attenuates S-nitrosylation of STING to facilitate its activation.**
**a** Immunoblot assays of S-nitrosylation of STING in PMs from *Adh5*⁺/⁺ or *Adh5*⁻/⁻ mice infected with HSV-1 using the IBP. **b** Immunoblot assays of dimer-STING in cGAMP-stimulated PMs from *Adh5*⁺/⁺ or *Adh5*⁻/⁻ mice. Immunoblot assays of p-TBK1, p-IRF3, and p-STAT1 in PMs from *Adh5*⁺/⁺ or *Adh5*⁻/⁻ mice stimulated with HSV-1 (**c**) or cGAMP (**d**). **e** ELISA analysis of IFN-β secretion in PMs from *Adh5*⁺/⁺ or *Adh5*⁻/⁻ mice, plus stimulation as indicated. **f** Immunoblot assays of p-TBK1, p-IRF3, and p-STAT1 in PMs transfected with negative control siRNA (si*NC*) or *Adh5* siRNA (si*Adh5-2*), followed by DMXAA stimulation. **g** qPCR analysis of *Ifnb* expression in PMs transfected with si*NC* or si*Adh5-2*, plus stimulation as indicated. **h** Immunoblot assays of p-IRF3 and p-STAT1 in PMs pretreated with increasing concentrations of N6022, followed by HSV-1 infection. **i** ELISA analysis of IFN-β secretion in PMs pretreated with N6022

and then stimulated with HSV-1 or DMXAA. **j** Luciferase activity analysis of IFN-β activation in HEK293T cells transfected with empty vector or cGAS plus STING plasmids, together with ADH5 plasmid or empty vector plasmid. *Adh5*⁺/⁺ or *Adh5*⁻/⁻ mice were stimulated with HSV-1 by intraperitoneal injection. Serum levels of IFN-β (**k**) and IL-6 (**l**) were analyzed by ELISA (PBS group, *n* = 1; HSV-1 group, *n* = 4). **m**, **n** *Adh5*⁺/⁺ or *Adh5*⁻/⁻ mice were stimulated with DMXAA by intraperitoneal injection. Serum levels of IFN-β were analyzed by ELISA (PBS group, *n* = 1; DMXAA group, *n* = 4) (**m**). qPCR analysis of *Ifnb* mRNA expression in spleen and lung tissues (**n**). Data represent mean ± SD or one representative from three independent experiments. The *p* values were calculated using unpaired two-sided *t* test and adjustments were made for multiple comparisons.

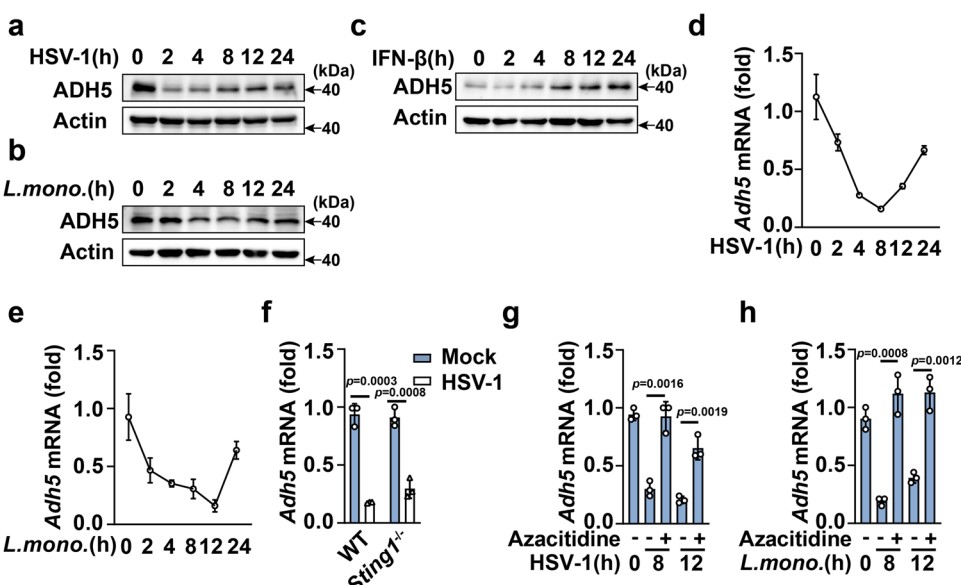

**Fig. 6 | ADH5 is downregulated during pathogens infection by promoting DNA methylation.** Immunoblot assays of ADH5 in PMs during HSV-1 (**a**), *L. monocytogenes* (**b**) infection or IFN-β (**c**) stimulation. qPCR analysis of *Adh5* mRNA expression during HSV-1 (**d**) or *L. monocytogenes* (**e**) infection. **f** qPCR analysis of *Adh5* mRNA expression in HSV-1-infected PMs from *Sting1*⁺/⁺ or *Sting1*⁻/⁻ mice. qPCR

analysis of *Adh5* mRNA expression in PMs pretreated with solvent (DMSO) or Azacitidine and then infected with HSV-1 (**g**) or *L. monocytogenes* (**h**). Data represent mean ± SD or one representative from three independent experiments. The *p* values were calculated using unpaired two-sided *t* test and adjustments were made for multiple comparisons.

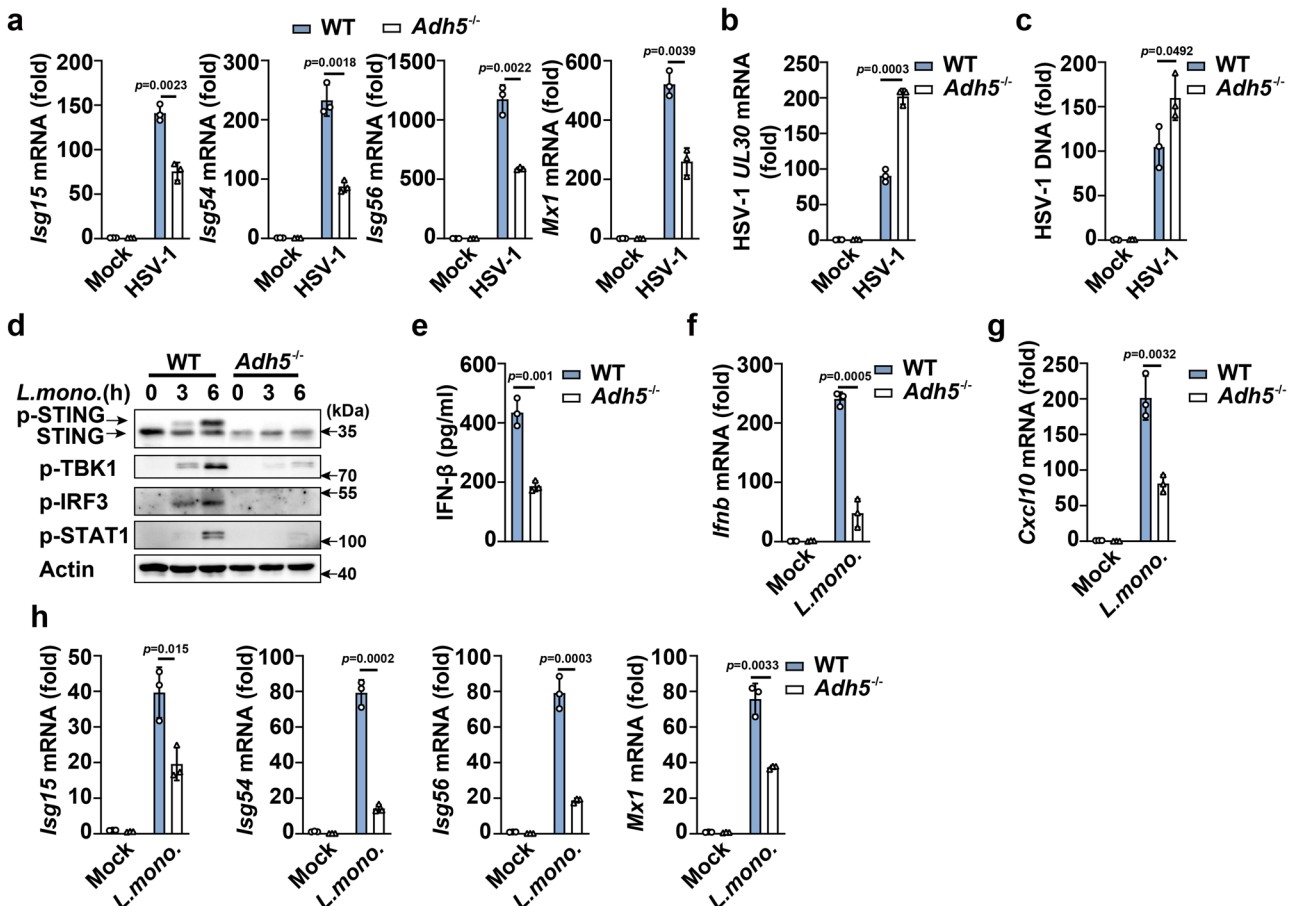

**Fig. 7 | ADH5 deficiency attenuates HSV-1- and *L. monocytogenes*-induced innate responses. a–c** qPCR analysis of *Isg15, Isg54, Isg56, Mx1*, HSV-1 *UL30* mRNA expression and HSV-1 DNA level in HSV-1-infected PMs from *Adh5⁺/⁺* or *Adh5⁻/⁻* mice. **d** Immunoblot assays of STING, p-TBK1, p-IRF3, and p-STAT1 in *L. monocytogenes*-infected PMs from *Adh5⁺/⁺* or *Adh5⁻/⁻* mice. **e** ELISA analysis of IFN-β secretion in *L.* *monocytogenes*-infected PMs from *Adh5⁺/⁺* or *Adh5⁻/⁻* mice. **f–h** qPCR analysis of *Ifnb, Cxcl10, Isg15, Isg54, Isg56*, and *Mx1* mRNA expression in *L. monocytogenes*-infected PMs from *Adh5⁺/⁺* or *Adh5⁻/⁻* mice. Data represent mean ± SD or one representative from three independent experiments. The *p* values were calculated using unpaired two-sided *t* test and adjustments were made for multiple comparisons.

An extensive set of PTMs control STING activity downstream of its binding to cGAMP. Palmitoylation of two cysteine residues (Cys88 and Cys91) facilitates STING oligomerization[48,49]. Polyubiquitination of STING, induced by several E3 ubiquitin ligases, including autocrine motility factor receptor, tripartite motif-containing protein (TRIM) 56, TRIM32, and mitochondrial ubiquitin ligase activator of nuclear factor-kappa beta 1, promotes STING trafficking and interacts with TBK1 to facilitate STING activation. Binding with cGAMP is the first and critical step in STING activation, which requires the formation of hydrogen bonds at several STING residues (Asn242, Ser241, Val239, Tyr163, Glu260, Tyr261, and Thr263)[5–7,9,22]. Our results indicate that GSNO induces S-nitrosylation of STING at Cys257, destroying the hydrogen bonds between cGAMP and E260/T263 residues of STING, further reducing the binding affinity of cGAMP to STING. Consequently, GSNO attenuated STING-triggered type I IFN responses in HSV-1 and *L. monocytogenes* infections. Furthermore, GSNO inhibited STING-triggered IL-6 secretion during HSV-1 infection, suggesting GSNO suppressed STING activation in a TBK1-independent manner. It should be noted that although the treatment with GSNO could not completely block STING-mediated innate immune responses in macrophages, mutant the S-nitrosylation site of STING most blocked cGAMP binding to STING, suggesting that there may be some resistance mechanism to prevent excessive S-nitrosylation in host immune cells. Therefore, SNO-induced S-nitrosylation blocks cGAMP binding and controls STING activity in the initial step of its activation.

The emergence of exogenous DNA or cyclic dinucleotides, such as c-di-AMP and c-di-GMP, in the cytoplasm during pathogen infection

activates STING to initiate host innate immune responses and facilitates the eradication of invading pathogens[5,50]. Pathogenic microorganisms have evolved sophisticated mechanisms to disable the immune system by targeting STING[51]. For example, viral infection causes the accumulation of nitro-fatty acids or fatty acid carbonyl products that covalently modify STING by nitro-alkylation or carbonylation, blocking its palmitoylation and activation[14,52]. Viral IRF1 in Kaposi sarcoma-associated herpesvirus and viral protein 1/2 in HSV-1 prevent STING activation by blocking the phosphorylation and K63-linked ubiquitination of STING, respectively[53,54]. In the present study, we showed that pathogen infection facilitated S-nitrosylation of STING to suppress host defense by downregulating ADH5 expression. *Adh5* deficiency triggers substantial increases in whole-cell S-nitrosylation during bacterial infection, resulting in enhanced tissue damage and lymphocyte apoptosis[18]. *Adh5* deficiency facilitates HSV-1 replication by suppressing host type I IFN response. In consideration of the pleomorphic effects of GSNO on protein S-nitrosylation, further understanding the role of ADH5 and GSNO in innate immune response in a STING-independent manner may contribute to fully understanding the relationship between oxidative burst and host defense. Therefore, ADH5 is downregulated during pathogens infection to circumvent STING-dependent type I IFN responses by modulating SNO homeostasis.

In summary, our study uncovered a novel mechanism that enables the activation of innate immune responses during an oxidative burst. We identified S-nitrosylation as a novel PTM of STING and clarified the

vital role of SNO homeostasis maintained by ADH5 in activating STING. As growing evidence implicates aberrant STING activity in various diseases[3,55], STING has emerged as an attractive target for pharmacological modulation. Our results suggest that modulation of SNO homeostasis by ADH5 is a promising strategy for the treatment of diseases caused by aberrant STING activation and expands our understanding of the functions of STING-dependent host defense, which could help the development of new strategies to modulate immune defense against pathogens.

## Methods

### Ethics statement

All animal experiments were performed in compliance with the National Institutes of Health Guide for the Care and Use of Laboratory Animals with approval from the Scientific Investigation Board of the School of Basic Medical Science, Shandong University, Jinan, Shandong Province, China. The approval number is LL-201602059.

### Mice and cells

*Adh5*-deficient (S-KO-00928) mice were generated by Cyagen Biosciences using CRISPR/Cas9-mediated genome editing. *Sting1*-deficient mice were obtained from Jackson Laboratory. Both male and female mice (C57BL/6 background) at 6–11 weeks of age were used. All mice were housed in a pathogen-free facility at the Model Animal Research Center of Shandong University. They were kept in day/night cycles (12 h each), with temperature of 20–26 °C and humidity 40–70%.

Mouse primary peritoneal macrophages (PMs) were isolated from mouse peritoneal cavities three days after intraperitoneal (i.p.) injection of 3% Brewer's thioglycollate. HEK293T and THP-1 cells were purchased from the American Type Culture Collection. 293-Dual hSTING-A162 cells, which were stably transfected with the Ala162 isoform of human STING (S162A) and confer sensitivity to DMXAA, were obtained from InvivoGen. BJ cells were purchased from Shanghai Fuheng Biotechnology Co., Ltd. For PMs, HEK293T, 293-Dual hSTING-A162, and BJ cells were cultured in Dulbecco's modified Eagle's medium supplemented with 10% fetal bovine serum, $100\,\mu$ ml$^{-1}$ penicillin, and $100\,\mu$g ml$^{-1}$ streptomycin. THP1 cells were cultured in RPMI 1640 medium under the same growth conditions. All cells were cultured at 37 °C in an atmosphere of 5% (v/v) $CO_2$. The Cell Counting Kit-8 (CCK-8) (APE×BIO) was used to quantitatively measure cell viability. The cell lines were checked for mycoplasma contamination using the MycoProbe detection kit (R&D Systems). Only the cells without mycoplasma contamination were used and authenticated by morphology, karyotyping, and polymerase chain reaction (PCR)-based approaches.

### Reagents and antibodies

GSNO (N4148), SNAP (N3398), and biotin-maleimide (B1267) were purchased from Sigma-Aldrich. DMXAA (S1537), N6022 (S7589), H2DCFDA (S9687) and Azacitidine (S1782) were obtained from Selleck Chemicals. The 2′3′-cGAMP (tlrl-nacga23-1) and c-di-GMP (tlrl-nacdg) was obtained from InvivoGen. HENS buffer (90106), Methyl methanethiosulfonate (MMTS) (23011), and streptavidin agarose (20349) were obtained from Thermo Fisher Scientific. Biotin-cGAMP (C157) was obtained from BioLog Life Science Institute. Stimulants were used at the following concentrations: GSNO, 200 μM; SNAP, 200 μM; N6022, 20 μM. Cells were pretreated with GSNO, SNAP or N6022 for 1 h. H2DCFDA, 10 μM; Azacitidine 10 μM; ISD, 5 μg ml$^{-1}$; cGAMP, 3 μg ml$^{-1}$; c-di-GMP, 5 μg ml$^{-1}$; and DMXAA, 150 μg ml$^{-1}$. HSV-1, HSV-GFP and *L. monocytogenes* were obtained from X. Cao (Second Military Medical University, Shanghai, China). The purified His-hSTING protein was purchased from aBIOTECH(Jinan, China).

Anti-cGAS (D3O8O, 31659), anti-STING (D1V5L, 50494), anti-p-IRF3 (Ser396, 4947), anti-p-STAT1 (Tyr701, 9167), anti-TBK1 (3013), anti-IRF3 (4302), and anti-biotin (D5A7,5571) antibodies were purchased from

Cell Signaling Technology. Anti-p-TBK1 (ab109272) and anti-GM130 (ab52649) antibodies were purchased from Abcam. Alexa Fluor 633 (A-21071) and 488 (A-11059) were purchased from Thermo Fisher Scientific. The anti-FLAG (F1804) antibody was purchased from Sigma-Aldrich. The anti-β-actin (66009-I-Ig) antibody was purchased from Proteintech. The anti-ADH5 (A13459) antibody was purchased from ABclonal. Anti-hSTING (MAB7169), used for immunofluorescence, was obtained from R&D Systems. ELISA kits were used to quantify IFN-β (BioLegend), IL-6 (Dakewe Biotech.) and cGAMP (Cayman Chemical) in the cellular supernatant or serum, according to the manufacturer's instructions.

### Plasmids, RNA interference and transfection

The C257S, C309S and R238A/Y240A STING mutants were generated using the KOD-Plus-Mutagenesis kit (Toyobo), and all constructs were confirmed by DNA sequencing. The ADH5-myc plasmid was purchased from Origene. The IFN-β reporter and expression plasmids for cGAS and STING have been previously described[14]. The plasmids were transiently transfected into HEK293T cells using Lipofectamine 2000 (Invitrogen). The target sequences in mouse *Adh5* used for transient silencing by small interfering RNA (siRNA) were 5′-CCCAGUGAUCUUGGGACAU-3′ (siRNA1), 5′-GGAUCAUUGGUAUCGACAU-3′ (siRNA2), and 5′-GCAUUC GAACUGUUCUAAA-3′ (siRNA3). The "scrambled" control sequence was 5′-UUCUCCGAACGUGUCACGU-3′. siRNA duplexes were transfected into PMs using INTERFERin reagent (Polyplus) according to the manufacturer's instructions.

### Reverse transcription real-time PCR (RT-PCR)

The RNAfast200 RNA Extraction kit (Fastagen) or TRIzol reagent (Invitrogen) was used to extract total RNA from cultured cells and tissues according to the manufacturer's instructions. Equal amounts of RNA were reverse-transcribed into complementary DNA (cDNA) using reverse transcriptase (Nanjing Vazyme Biotech Co.) The cDNAs were subjected to quantitative RT-PCR (qRT-PCR) analysis using SYBR Green (Nanjing Vazyme Biotech Co.). The primers used to detect the expression of the indicated RNAs are listed in Supplementary Table 1.

### Extraction of viral DNA

DNA was extracted from HSV-1 infected cells using TIANamp Genomic DNA kit (TIANGEN) according to the manufacturer's instructions. Quantitative real-time polymerase chain reaction (PCR) was subsequently conducted for quantifying HSV-1 DNA.

### Immunoblot analysis

Cells were washed with phosphate-buffered saline (PBS) and lysed using RIPA Protein Extraction Reagent (Pierce, Thermo Fisher Scientific) supplemented with a protease inhibitor cocktail (Sigma-Aldrich) and phosphatase inhibitor cocktail (CoWin BioSciences). A bicinchoninic acid protein assay kit (Pierce, Thermo Fisher Scientific) was used to determine the protein concentrations in the extracts. The adjusted lysates were loaded onto and subjected to 10% sodium dodecyl sulfate-polyacrylamide gel electrophoresis (SDS-PAGE). The proteins were then transferred onto nitrocellulose blotting membranes (Millipore) for immunoblot analysis as previously described[14].

### Viral infection in vivo

C57BL/6 mice (seven-week-old females) were infected by i.p. ($2 \times 10^7$ plaque-forming units per mouse) or i.v. ($1 \times 10^8$ plaque-forming units per mouse) injection of HSV-1. GSNO (1 mg kg$^{-1}$) was administered by i.p. injection for three days before viral infection. For wild-type (WT) and *Adh5*-deficient mice, seven-week-old sex-matched littermates were intraperitoneally administered DMXAA (12.5 mg/kg body weight). Serum was collected from the mice for ELISA 8 h after HSV-1 infection or 6 h following DMXAA stimulation. Brain, spleen, and lung tissues were collected for qRT-PCR analysis. Lungs from mice infected with HSV-1 for 36 h were dissected for hematoxylin-eosin staining and examined by

light microscopy for histological changes and by RT-PCR for HSV-1 replication. For plaque assays, mice were intravenously injected with HSV-1 ($1^{\wedge}10^8$ PFU per mouse) and the lung were collected and homogenized in DMEM at day 2 after infection. For HSV-1 infection survival experiments, mice were treated with GSNO three days before infection with HSV-1 and then monitored for survival after viral infection.

### In vitro pull-down assay

PMs or HEK293T cells were lysed in NP-40 lysis buffer, followed by centrifugation at 13,000 g for 15 min. Supernatants were incubated with biotin-cGAMP (BioLog Life Science Institute) at 4 °C for 4 h while rotating and then incubated with 20 μL of streptavidin beads at 4 °C for another 4 h. The beads were washed four times with lysis buffer and subjected to immunoblotting to analyze STING binding.

### STING dimerization assay

PMs were lysed using RIPA Protein Extraction Reagent (Pierce, Thermo Fisher Scientific) supplemented with a protease inhibitor cocktail (Sigma-Aldrich), followed by centrifugation at 13,000 g for 15 min. Non-reducing SDS-PAGE sample loading buffer was added to the supernatants without boiling and then subjected to immunoblotting.

### STING oligomerization assay

Cells were lysed in RIPA buffer with a protease inhibitor cocktail. After centrifugation, the cell lysates were mixed with loading buffer without boiling. Electrophoresis was performed on ice under low voltage.

### Immunofluorescence and confocal microscopy analyses

BJ cells were plated on glass coverslips in 24-well plates. The BJ cells were pretreated with GSNO for 1 h followed by cGAMP delivery by permeabilization with digitonin (10 μg ml⁻¹) for 15 min in buffer (50 mM HEPES, pH 7.2, 100 mM KCl, 3 mM MgCl₂, 0.1 mM dithiothreitol [DTT], 85 mM sucrose, 0.2% bovine serum albumin [BSA], 1 mM adenosine triphosphate, and 0.1 mM guanosine triphosphate). For immunofluorescence analysis, BJ cells were fixed with Immunol staining fix solution (Beyotime), permeabilized with 0.5% Triton-X 100 in PBS, and blocked with 3% BSA for 1 h. GM130 and STING were labeled using the respective primary antibodies overnight at 4 °C. Subsequently, the secondary antibody (Alexa Fluor 633 or 488) was added for 1 h. Lastly, the cells were subjected to microscopy analysis using a Zeiss LSM880 confocal laser microscope provided by the Micro Characterization Facility at Shandong University. PMs were pretreated with GSNO followed by HSV-GFP infection for 12 h. After washing 3 times with phosphate-buffered saline (PBS), fluorescence was measured using a fluorescence microscope.

### Detection of S-nitrosylation by irreversible biotinylation procedure (IBP)

IBP was performed as previously described[22]. For PMs, cells were lysed using RIPA buffer to detect endogenous S-nitrosylation by IBP. HEK293T cells were co-transfected with plasmids for 24 h, lysed using RIPA buffer, and then the supernatant was precipitated by ice-cold acetone, and the pellet was recovered in HEN buffer with or without 1 mM GSNO for 1 h at room temperature. The purified His-hSTING protein was also treated with or without 1 mM GSNO for 1 h at room temperature. After removing GSNO by precipitation with cold acetone and centrifugation from the protein solution of HEK293T cells or purified His-hSTING protein, the pellet was recovered in HENS buffer containing 20 mM methyl methanethiosulfonate (MMTS) and incubated at 50 °C for 30 min to block free thiols. After ice-cold acetone precipitation, the pellet was resuspended in HENS buffer containing 0.2 mM biotin-maleimide with 10 mM ascorbate and incubated at 37 °C for 1 h. Excess biotin-maleimide was removed by acetone precipitation for 20 min, followed by centrifugation at 2000 g for 10 min. The samples were recovered and boiled in HENS buffer containing 200 mM

DTT for 15 min to reduce potential intermolecular disulfide bonds. Except for STING human recombinant protein, which directly labeled the S-nitrosylated STING with the biotin antibody, the biotinylated protein was purified using 20 μL streptavidin agarose beads by incubating with rotation at 4 °C for 2 h. After washing the beads thrice with HENS buffer, the samples were analyzed using western blotting to detect S-nitrosylated STING.

### Analysis of S-nitrosylated sites of STING using LC-MS/MS

The purified His-hSTING protein (200 μg) was treated with 1 mM GSNO for 1 h at room temperature. Gel pieces containing S-nitrosylated proteins were decolorized, washed until they were transparent, and dehydrated using acetonitrile. Proteins were alkylated using iodoacetamide and incubated for 45 min in the dark at 25 °C. The flow-through was discarded. Subsequently, 100 μl UA was added and centrifuged at 14,000 g for 15 min; this step was repeated once. Approximately 200 μl of 50 mM ammonium bicarbonate (ABC) was added and centrifuged twice at 14,000 g for 15 min. In a new Eppendorf tube, 4 μg of trypsin was then added and incubated at 37 °C for over 16 h. The samples were centrifuged at 14,000 g for 10 min, and the flow-through was collected in a new collection tube. Subsequently, 50 mM of ABC was added and centrifuged at 14,000 g for 10 min, and the flow-through was collected and dried using a SpeedVac. The sample was dissolved in 0.1% trifluoroacetic acid, desalted using C18 ZipTips, and dried using a SpeedVac. The samples were then resuspended in 0.1% formic acid for MS. Peptide samples were analyzed using an LTQ Orbitrap ETD mass spectrometer (Thermo Fisher Scientific). Briefly, the samples were loaded onto a high-performance liquid chromatography system (Easy-nLC 1000; Thermo Fisher Scientific) equipped with a C18 column (1.8 mm, 0.15 × 1.00 mm). Solvent A contained 0.1% formic acid, and solvent B contained 100% acetonitrile at a flow rate of 600 nL/min. MS analysis was performed at AIMSMASS Co., Ltd. (Shanghai, China) in the positive-ion mode with an automated data-dependent MS/MS analysis with full scans (350–1600 m/z) acquired using Fourier transform MS at a mass resolution of 30,000. The ten most intense precursor ions were selected for MS/MS. MS/MS was performed using higher-energy collision dissociation at 35% collision energy at a mass resolution of 15,000. Raw MS files were analyzed using Proteome Discoverer 1.4 (Thermo Fisher Scientific), and the parameters used for data analysis included trypsin as the protease with a maximum of two missed cleavages. The mass tolerances for precursor and fragment ions were set to 20 ppm and 0.05 Da, respectively. The search included variable modifications of oxidation, deamidation, and S-nitrosylation.

### Molecular dynamics (MD) simulation

The MD simulation was adapted from the GROMACS 2018.4 program[56] with constant temperature, pressure, and periodic boundary conditions using the amber14SB full atomic force field and transferable intermolecular potential with 3 points water model. In the MD simulation, all bonds involving hydrogen atoms were constrained using the LinCS algorithm, and the integration step was 2 fs. The electrostatic interactions were calculated using the particle mesh Ewald method. The truncation value of the non-bonded interaction was set as 10 Å and updated every 10 steps. The V-rescale[7] temperature coupling method was used to control the simulation temperature at 298.15 K, and the Parrinello–Rahman method was used to control the pressure at 1 bar. First, the steepest descent method was used to minimize the energy of the two systems to eliminate close contact between atoms; then, 1 ns NVT (constant volume, constant temperature) and NPT (constant pressure, constant temperature) equilibrium simulations were performed at 298.15 K. Finally, a 100 ns MD simulation of the system was performed, and the conformation was saved every 10 ps. The simulation results were completed and visualized using the GROMACS embedded program and Visual Molecular Dynamics.

## Luciferase activity assay

HEK293T cells were transiently co-transfected with plasmids using the Jet-PEI transfection reagent (Polyplus) for 24 h. The dual-luciferase reporter assay system (Promega) was used for the luciferase activity assay, according to the manufacturer's instructions. To determine the transfection efficiency, the data were normalized by dividing firefly luciferase activity by that of Renilla luciferase. For 293T-Dual hSTING-A162 cells, GSNO or SNAP was added to each sample well in a flat-bottom 96-well plate 1 h in advance, and then DMXAA was added to each well. Subsequently, 10–20 μl of cell culture supernatant was added to each well of a 96-well white plate. Lastly, 50 μl of QUANTI-Luc™ was added to each well to measure luciferase activity.

## Griess assay

Nitric oxide (NO) secretion was assessed using the Griess reagent (Beyotime Biotechnology, S0021) according to the manufacturer's instructions.

## Flow cytometry analysis

The production of ROS in cells was measured using the H2DCFDA probe (S9687, Selleck, USA) according to the manufacturer's recommendation. The stained cells were detected using a CytoFLEX cytometer instrument (Beckman Coulter, Brea, CA, USA) and analyzed using FlowJo v10.8 software. Dead and adherent cells were discriminated out by gating on live/dead population.

## Statistical analysis

All experiments were repeated at least thrice. Data are presented as mean ± standard deviation (SD). All quantitative measurements were tested for normal distribution. Comparisons between two groups were performed using an unpaired two-tailed Student's $t$ test. $P$ value was corrected for multiple comparisons using the Holm-Sidak method. Survival curves were compared using the Kaplan–Meier survival method. Statistical analyses were performed using the GraphPad Prism 9 software.

## Reporting summary

Further information on research design is available in the Nature Portfolio Reporting Summary linked to this article.

## Data availability

All data supporting the findings of this study are available within the article and its supplementary information files. They may also be obtained from the corresponding author upon reasonable request. Source data are provided with this paper.

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

## Acknowledgements

This work was supported by grants from the National Natural Science Foundation of China (grant no. 82125020 and 31870866 to W.Z. grant no. 82101855 to M.J. and grant no. 82321002 to C.G.) and the Postdoctoral Science Foundation of China (grant no. 2021T140406 to M.J.). We thank Translational Medicine Core Facility of Shandong University for the consultation and instrument availability that supported this work and Sichuan MoDe Technology for the technical support of molecular dynamics simulation.

## Author contributions

W.Z. and M.J. conceived the project, designed the experiments, analyzed the data, and wrote the manuscript. M.J., L.C., J.W., and M.W. performed most of the experiments. D.Q., H.S., Y. F., and C.Z. assisted with experiments and provided technical assistance. C. G. and J.J. provided the expertize and advice. W.Z. provided the overall direction.

## Competing interests

The authors declare no competing interests.
