## [Peer Review File · Nature Communications]

S-nitrosothiol homeostasis maintained by ADH5 facilitates STING-dependent host defense against pathogensEditorial Note: This manuscript has been previously reviewed at another journal that is not operating a transparent peer review scheme. This document only contains reviewer comments and rebuttal letters for versions considered at *Nature Communications*.

REVIEWERS' COMMENTS

Reviewer #1 (Remarks to the Author):

I thank the authors for properly addressing my previous concerns.

Reviewer #2 (Remarks to the Author):

The authors have satisfactorily addressed my previous points.

Reviewer #3 (Remarks to the Author):

The authors find that treatment with S-nitrosoglutathione (GSNO), which is a reactive S-nitrosothiol (SNO), inhibits cGAS-STING activation, reducing the expression of host-defense genes, and sensitizing mice to HSV-1 infection. They find that GSNO treatment induces STING S-nitrosylation at C257, which prevents cGAMP binding, STING dimerization/oligomerization, and subsequent cGAS/STING downstream signaling. Further, they find that ADH5, which is an S-nitrosoglutathione reductase, is necessary to prevent STING nitrosylation and maintain normal cGAS-STING signaling and transcriptional activation. These results highlight how reactive nitrogen species can mechanistically modulate a critical host defense pathway. This raises interesting questions about the extent to which cells and intracellular pathogens modulate RNS levels and signaling, and how these modifications are reversed or maintained to modulate host defense. These results also raise interesting possibilities that nitrosylation of STING could be relevant in other biologically important contexts.

Limitations of the article

-The effect of GSNO on cGAMP treatment or infection-mediated host defense gene expression is relatively small compared to the full induction in control samples. For example, various genes were induced ~100-500 fold relative to non-treated cells. Treatment with GSNO reduced this induction by about 50%, i.e. there was still a ~50-250-fold induction in gene expression (Fig 1). This implies a large amount of STING is still active despite treatment with high concentrations of GSNO. It would be helpful for readers to acknowledge this in the discussion.

-The pleomorphic effects of GSNO on protein nitrosylation, as well as those associated with ADH5 inactivation on SNO levels, make it difficult to definitively ascribe the observed host-defense phenotypes to STING nitrosylation specifically. It would be helpful to raise this caveat in the discussion.

Reviewer #4 (Remarks to the Author):

The authors have done extensive work to respond to a number of reviewer concerns and I think this has significantly improved the manuscript, most notably really demonstrating the specificity of this response for STING and not IFN induction more generally. While the vast majority of my concerns have been addressed I still believe the authors are overstating their findings by saying that "Pathogens downregulate ADH5 to attenuate immune responses". This suggests an active mechanism mediated by specific factors from the pathogens. There is no evidence provided for this in the manuscript. While the authors convincingly show that this happens during infection, it is not clear if this is host or pathogen driven and without a specific pathogen mechanism provided I feel strongly that this unfounded interpretation needs to be removed from the manuscript. The authors demonstrate that infection drives DNA methylation but not that this is a pathogen driven process.

Reviewer #1:

I thank the authors for properly addressing my previous concerns.

Answer: We appreciate very much for your work in reviewing our manuscript. The insightful suggestions and comments help us to greatly improve the manuscript.

Reviewer #2:

The authors have satisfactorily addressed my previous points.

Answer: We appreciate very much for your work in reviewing our manuscript. The insightful suggestions and comments help us to greatly improve the manuscript.

Reviewer #3:

The authors find that treatment with S-nitrosoglutathione (GSNO), which is a reactive S-nitrosothiol (SNO), inhibits cGAS-STING activation, reducing the expression of host-defense genes, and sensitizing mice to HSV-1 infection. They find that GSNO treatment induces STING S-nitrosylation at C257, which prevents cGAMP binding, STING dimerization/oligomerization, and subsequent cGAS/STING downstream signaling. Further, they find that ADH5, which is an S-nitrosoglutathione reductase, is necessary to prevent STING nitrosylation and maintain normal cGAS-STING signaling and transcriptional activation. These results highlight how reactive nitrogen species can mechanistically modulate a critical host defense pathway. This raises interesting questions about the extent to which cells and intracellular pathogens modulate RNS levels and signaling, and how these modifications are reversed or maintained to modulate host defense. These results also raise interesting possibilities that nitrosylation of STING could be relevant in other biologically important contexts.

Answer: We appreciate very much for your work in reviewing our manuscript. The insightful suggestions and comments help us to greatly improve the manuscript. We hope that the correction will meet with approval.

Limitations of the article

-The effect of GSNO on cGAMP treatment or infection-mediated host defense gene expression is relatively small compared to the full induction in control samples. For example, various genes were induced ~100-500 fold relative to non-treated cells. Treatment with GSNO reduced this induction by about 50%, i.e. there was still a ~50-250-fold induction in gene expression (Fig 1). This implies a large amount of STING is still active despite treatment with high concentrations of GSNO. It would be helpful for readers to acknowledge this in the discussion.

Answer: Thanks for the important comments. We have discussed in the part of “Discussion” as follows: “It should be noted that although the treatment with GSNO could not completely block STING-mediated innate immune responses in macrophages, mutant the S-nitrosylation site of STING most blocked cGAMP binding to STING, suggesting that there may be some resistance mechanism to prevent excessive S-nitrosylation in host immune cells.”

-The pleomorphic effects of GSNO on protein nitrosylation, as well as those associated with ADH5 inactivation on SNO levels, make it difficult to definitively ascribe the observed host-defense phenotypes to STING nitrosylation specifically. It would be helpful to raise this caveat in the discussion.

Answer: Thanks for the important comments. We have discussed in the part of “Discussion” as follows: “In consideration of the pleomorphic effects of GSNO on protein S-nitrosylation, further understanding the role of ADH5 and GSNO in innate immune response in a STING-independent manner may contribute to fully understanding the relationship between oxidative burst and host defense.”

Reviewer #4:

The authors have done extensive work to respond to a number of reviewer concerns and I think this has significantly improved the manuscript, most notably really demonstrating the specificity of this response for STING and not IFN induction more generally. While the vast majority of my concerns have been addressed I still believe the authors are overstating their

findings by saying that "Pathogens downregulate ADH5 to attenuate immune responses". This suggests an active mechanism mediated by specific factors from the pathogens. There is no evidence provided for this in the manuscript. While the authors convincingly show that this happens during infection, it is not clear if this is host or pathogen driven and without a specific pathogen mechanism provided I feel strongly that this unfounded interpretation needs to be removed from the manuscript. The authors demonstrate that infection drives DNA methylation but not that this is a pathogen driven process.

Answer: We appreciate very much for your work in reviewing our manuscript. The insightful suggestions and comments help us to greatly improve the manuscript.

We have corrected "Pathogens downregulate ADH5 to attenuate host innate responses" as "ADH5 is downregulated during pathogens infection that attenuates host innate responses" in the part of "Results" and "Discussion".